# Is There Urban Landscape in Metropolitan Areas? An Unobvious Answer Based on Corine Land Cover Analyses

**Urszula Myga-Piątek** *[ID], **Anna Żemła-Siesicka** [ID], **Katarzyna Pukowiec-Kurda, Michał Sobala** [ID] **and Jerzy Nita**

Faculty of Natural Sciences, University of Silesia, 41200 Sosnowiec, Poland;
anna.zemla-siesicka@us.edu.pl (A.Ż.-S.); katarzyna.pukowiec@us.edu.pl (K.P.-K.);
michal.sobala@us.edu.pl (M.S.); jerzy.nita@us.edu.pl (J.N.)
\* Correspondence: urszula.myga-piatek@us.edu.pl; Tel.: +48-32-3689-361

**Abstract:** The recent increase in urban areas has stimulated landscape urbanization. One of the ways to study this process is an analysis based on the structure of land cover. The aim of this paper is to assess the intensity of the urban landscape on the basis of the CORINE in the seven largest metropolitan areas in Poland and in the Ruhr Metropolis in Germany. To this end, an urban landscape intensity indicator (ULII) was used based on Corine Land Cover at three levels of detail: the metropolitan area, municipalities and hexagons. There are similarities in landscape structure in areas with similar origin (industrial function) and spatial organization (mono- and polycentric agglomerations). The landscape of the Upper Silesia-Zagłębie Metropolis differs from the landscape of other metropolitan areas in Poland and simultaneously shows similarities to the landscape of the Ruhr Metropolis. The results of the ULII also revealed a dependency: the dominance of rural and transitional landscapes in a majority of the study areas. Urban landscapes occur only in the central zones of the metropolitan areas. This proves that determining the range of a metropolitan area in terms of landscape factors is different from doing it with formal or legal ones.

**Keywords:** landscape urbanization; metropolises; agglomeration in Poland; urban landscape intensity index

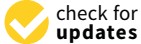



## 1. Introduction

Urbanization has accompanied humankind since antiquity and is a fundamental feature in every civilization. Urbanization refers both to built-up agglomerations and ways of life and describes the process of rural areas transforming into urban ones [1]. During the period of industrialization in the nineteenth century, urbanization processes began to acquire a fast rate and, since then, have become a common phenomenon in expanding cities [2]. The increase in the number of people living in urban areas means that in some European countries this phenomenon affects almost 80% of the population [3]. In addition, some researchers say that by 2030, 75% of all Europe's landscapes will be urban in nature [4]. The constant growth of urban areas is visible mainly around areas of large agglomerations and is related to the processes of suburbanization and urban sprawl [5,6].

The above processes change the structure of the landscape. In this paper, landscape is understood as a combination of different types of land cover [7]. Thus, along with an increase in the degree of urbanization of the landscape, urban areas increase and natural, rural and/or transitional areas decreases. The processes of landscape urbanization have a fundamental impact on its structure and physiognomy and depend on the intensity of changes and technological progress. Thus, in the cities one can speak of an urban landscape, but is this really evident? An urban landscape is understood to be an existing landscape of urban settlements and their surroundings, but with marked types of use of the city and without administrative restrictions [8]. The authors understand the urbanization of landscape as the process of differentiation and concentration of the internal spatial structure of cities and the quantitative increase in the area of landscapes of urban types. This is a

result of spatial expansion (urban sprawl) or the transformation of other types of landscapes (suburban, agricultural, forest, etc.) into an urban one. This process is always associated with an increased human impact (or an increase in the gradient of anthropopressure). One important aspect of the urban landscape is its complexity and multidimensionality, which means the living environment of the societal, cultural, historical and economic contexts and a matrix for further development [9].

At the same time as constant socio-economic development, some cities can gain a special hierarchical position in relation to other cities, transforming into metropolises [10]. Therefore, one of the possible ways for the urban landscape to evolve is through the growth of the metropolitan landscape [11]. Metropolises are a unique type of urban complex. Due to the dynamic processes of not only urbanization but also metropolization, the concept of metropolitan landscape is more often considered in landscape geography and in landscape architecture to be a unique type of urban landscape [12]. The concept of metropolitan landscape is related to the fact that the terms "city" and "urban" landscape are inadequate in describing the fullness and enormity of the roles played by urban networks, spaces, environments and processes. They extend far beyond the city area to the peripheral areas around it, both rural and urban [13]. Beunen [14] believes that metropolitan landscape is like an urban field encompassing built-up and open areas within urban centers.

The urban landscape is characterized by the complexity of its aspects, but also by the dynamics resulting from continuous transformation processes [15]. In Poland, the processes of landscape metropolization are in the initial stages; therefore, it is first necessary to recognize the directions of change and the dynamics of urban landscapes. Taking into account the complexity of the concept of urban landscape, it can be considered from many angles and at many research levels [3]. In terms of geography, it means the spatial organization of its individual elements, which are patches of land cover. Their analysis enables the examination of the landscape structure and its identification.

There are various methods in the literature for determining the degree of landscape urbanization. These are mostly based on land cover analysis and land use change. These methods have been tested in various regions, aspects and spatial relationships by many researchers. In the area of Central Europe, studies on the degree of urbanization were carried out by Inostroza et al. [16]. The aim of their work was to perform a spatially explicit quantification of urbanization degrees across the landscape. Dadashpor et al. [17] analyzed land cover changes in the Tabiz metropolitan area in Azerbaijan using the dispersion of urban lands as the landscape indicator. Changes in land cover due to urbanization were also analyzed by Aguilera et al. [18] and Reis et al. [19] using landscape metrics to analyze urban land cover in expanding cities. Tate et al. [20] assessed the impact of urbanization on the natural environment using the multimetric urban intensity index based on variables related to land cover as well as its management and socio-economic status. A similar set of variables was used by McMahon et al. [21] who examined the relationship between the degree of urbanization and the quality of surface waters in areas with high dynamics of urban processes. The relationship between urbanization processes and type of land cover was studied by Li et al. [22] in the Beijing area and Weng [23], who analyzed the urban gradient in Wisconsin. Huang et al. [24] dealt with the problem of converting rural landscapes into urban ones based on land cover analysis using statistical methods—the regression model. Fuzzy set theory was used to identify the transition zone between the urban and rural landscape in Olsztyn and Sieradz (Poland) [25,26]. The types of coverage analyzed on the basis of satellite images allowed trends in the urbanization of Eastern European cities to be identified [27]. The landscape of some metropolitan areas in Poland was assessed, e.g., Trójmiasto (Gdańsk-Gdynia-Sopot, Marciniak) and Poznań [28,29].

The latest research by Naranjo Gómez et al. [30] concerned the analysis of land cover types and changes that took place in the Canary Islands. In this work, the CORINE land cover was used as the basic research material. The CLC database consists of an inventory of land cover in 44 classes. It covers 39 countries, comprising the European Environment Agency (EEA) members and cooperating countries, including the members of the European

Union. This fact is crucial if comparing of land cover in different countries is the aim of the study. The CLC database supports broad spatial analyses because the data describing land cover in Europe are characterized by spatial continuity and enable non-ambiguous identification of various land-use types. Cieślak et al. [31] also used the same database in their analyses to assess the urban sprawl process. Similar research was carried out in by Solecka et al. [32], who assessed this process in the suburban area of Wrocław on the basis of the CORINE database. Benito et al. [33] also based their research on the CORINE model in determining land cover changes in the Mediterranean area. Research on the relationship between urban pattern and land cover was conducted in the Toronto agglomeration. For this purpose, researchers used the NDVI index as well as the urban patterns and socio-economic variables [34].

The aim of the article is to assess the intensity of the urban landscape on the basis of the CORINE database for seven metropolitan areas in Poland and one in Germany. To this purpose, the Urban Landscape Index (ULII) was introduced according to the formula presented by Matuszewska and Będkowski [26]. This index is based on the classification of land cover types in terms of their urban, transitional and rural character proposed by Biłozor [35]. The analyses were conducted at three levels of research details and comparisons made across the cities. The intensity of the urban landscape was calculated at the level of entire metropolitan areas, for administrative units (municipalities) and basic fields in the form of hexagons with an area of 4 km$^2$.

In this paper, we consider a metropolitan area as an area established based on administrative decisions. Simultaneously, it is an area that aspires to be a metropolis in the future. The pace of development of metropolis depends on many factors, but one of them is the presence of urban landscapes, surrounded by non-urban landscapes. Delineating of metropolitan areas in Poland is based on administrative decisions. These decisions take into account varied criteria, such as functional and economical. Their importance is currently discussed in the scientific literature. Landscape criteria have not been taken into account in these decisions so far. It should be expected that delineating metropolitan areas using landscape criterion would result in a completely different spatial extent of metropolitan areas. Since the administratively established metropolitan areas in Poland are characterized by the presence of urban, rural, and urban–rural municipalities, it can be assumed that non-urban landscapes will also occur within the metropolitan areas. It seems to be a paradox. Hence, the following questions may be posed: how many urban landscapes are in metropolitan areas, and how much city is in a city?

In Poland, only one of metropolitan areas is a polycentric agglomeration (Upper Silesia-Zagłębie Metropolis—US-ZM), while the rest are monocentric agglomerations. Presumably, the US-ZM should be distinguished in terms of landscape due to both the polycentric form of agglomeration and the matter of its origins, because it is the only metropolitan area in Poland whose formation is associated with the mining industry. For the sake of comparison, the Ruhr Metropolis was analyzed, assuming that due to the same genesis, the area might be similar in terms of landscape to the US-ZM.

The metropolitan areas in Poland were not compared in terms of urban landscape so far, although they were studied in terms of agriculture [36]. Additionally, the topic of theirs administrative and landscape dissociation was not raised. The need for landscape studies assessing the intensity of the urban landscape results from the dynamics of processes related to urbanization (suburbanization, urban sprawl, metropolization). The dynamism of these processes generates the need to monitor the areas subject to them.

## 2. Materials and Methods

The research procedure can be divided into several basic stages (Figure 1). The first step of the research was the selection of the metropolitan areas. According to the classification of urban centers in Poland [37], the largest seven metropolitan areas in Poland were chosen based on the criterion of population. The other criterion of comparison was the origins of metropolitan area, thus the Ruhr Metropolis (RM) in Germany was chosen.

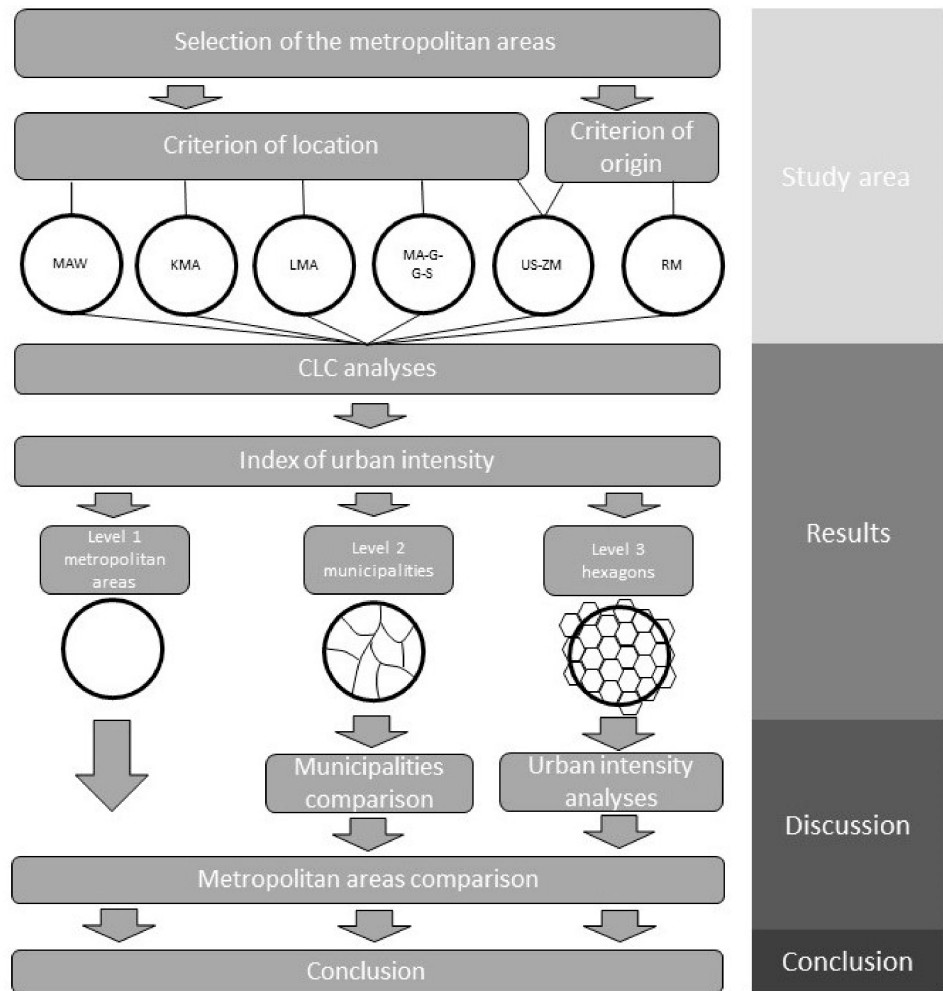

**Figure 1.** Schematic diagram of the research procedure. Symbols of the metropolitan areas—see Table 1.

Next, the delimitation and classification of land cover types in terms of their urban character were provided. For each land cover type, a value M (the degree of belonging to the urban landscape) was established, showing the urban–rural character of the type. Next, the urban landscape intensity index (ULII) was calculated on three levels of research (top-down approach): ULII value for each metropolitan area (to compare the degree of urban intensity of all studied areas), for each municipality within metropolitan areas (to shows the spatial differentiation and structure of landscape in each metropolitan area), and for geometric basic units—hexagons (to indicate regularities in the metropolitan areas in more detail). After obtaining the index, all metropolitan areas were compared in terms of the spatial distribution of urban, rural and transitional landscapes.

### 2.1. Study Area

The study area includes: The Metropolitan Area of Warsaw (MAW), the Upper Silesia-Zagłębie Metropolis (US-ZM), the Kraków Metropolitan Area (KMA), the Poznan Metropolitan Area (PMA), the Metropolitan Area of Gdańsk-Gdynia-Sopot (MAG-G-S), the Wrocław Metropolitan Association (WMA) and the Łódź Metropolitan Area (LMA) and the Ruhr Metropolis (RM, Germany) (Figure 2, Table 1). The majority of the studied metropolitan areas are monocentric agglomerations. Only US-ZM and RM have a different spatial character (polycentric). Furthermore, both US-ZM and RM have an industrial (mining) origin, similar polycentric character and high population density (the value of the population density index: 904 and 1163) [38].

**Table 1.** Basic characteristics of analyzed metropolitan areas.

| Symbol | Name | Legal Act/Document | Population [mln] | Number of Municipalities | Area [km²] |
|---|---|---|---|---|---|
| MAW | Metropolitan Area of Warsaw | development strategy | 3.08 [1] | 72 | 6206 |
| US-ZM | Upper Silesia-Zagłębie Metropolis | legally established metropolitan association | 2.30 [2] | 41 | 2545 |
| KMA | Kraków Metropolitan Area | spatial development plan | 1.51 [3] | 51 | 4060 |
| PMA | Poznań Metropolitan Area | spatial development plan | 1.33 [4] | 45 | 6198 |
| MAG-G-S | Metropolitan Area Gdańsk-Gdynia-Sopot | Municipalities association, draft act | 1.50 [5] | 56 | 6667 |
| WMA | Wrocław Metropolitan Association | spatial development plan/municipalities association | 1.20 [6] | 44 | 6719 |
| LMA | Łódź Metropolitan Area | development strategy/spatial development plan | 1.10 [7] | 28 | 2496 |
| RM | Ruhr Metropolis | Municipalities association | 5.10 [8] | 33 | 4385 |

[1] Strategy of development of the Metropolitan Area of Warsaw until 2030, 2015. [2] https://metropoliagzm.pl. [3] Krakow Metropolitan Area in 2011–2015, 2016. [4] Delimitation of Poznan Metropolitan Area, WBPP (http://www.wbpp.poznan.pl/opracowania/POM/POM.html). [5] www.metropoliagdansk.pl. [6] Spatial development plan of Lower Silesian voivodship. [7] Strategy of development of Łódź Metropolitan Area 2020+. [8] https://metropole.ruhr.

In this article, administrative criterion of delimitation of metropolitan area was adapted, based on diverseofficial documents (Table 1) [39–46]. The borders adopted in official documents delimit very large areas. It must be emphasized that according to these administrative bases, metropolitan areas may also include rural communities. The structure of the types of communities in Polish metropolitan areas is presented in Figure 3. Analyzing this characteristic, only in US-ZM in Poland urban municipalities dominates.

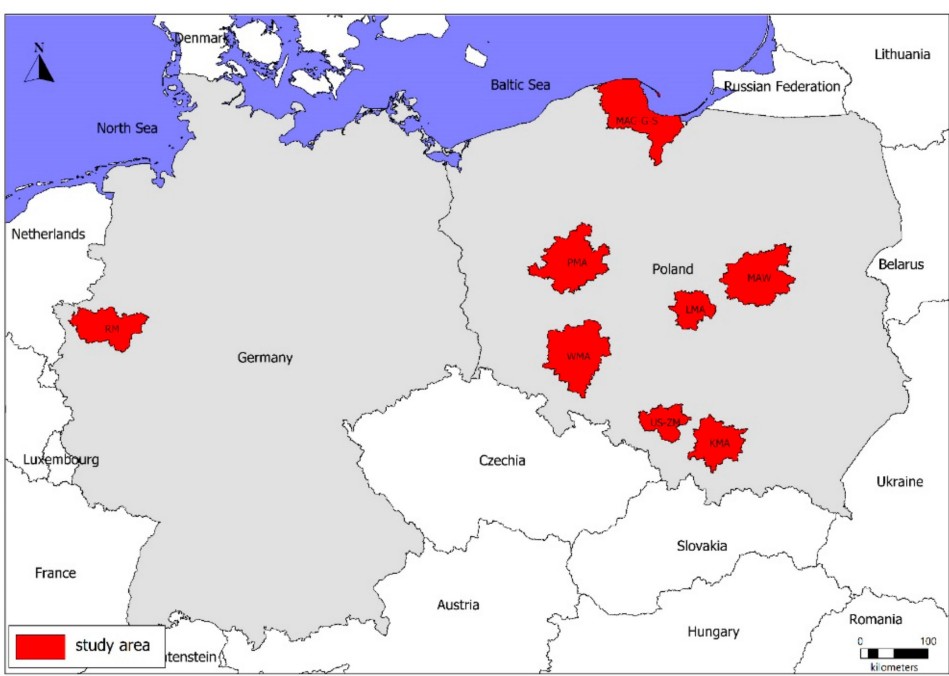

**Figure 2.** Study area. Symbols of the metropolitan areas—see Table 1.

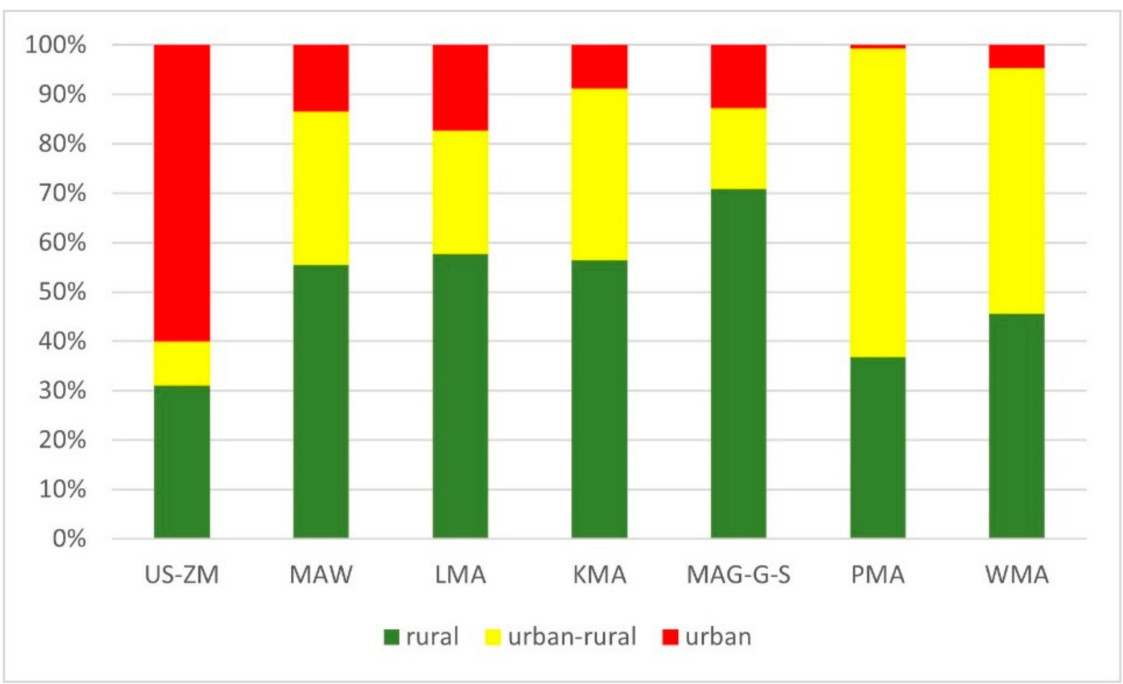

**Figure 3.** Percentage of administrative types of communities in Polish metropolitan areas.

*2.2. Materials*

In the presented study, the Corine Land Cover (CLC) database from 2018, Chief Geodesy and Cartography Office (GUGiK), and OpenGeodata of Nordrhein-West walen data were used. The analyses were conducted using the MapInfo Pro 17.0.

It must be emphasized that studies relying on CLC data have certain limitations, such as the detailed nature of the input data and a high degree of generalization. The CLC is a far more useful resource for small-scale studies, but it is a less reliable tool for analyses conducted on a larger scale [47].

The CLC has its own nomenclature that is consistent over the databases from different years and for the whole area. The nomenclature consists of 3 hierarchical levels, which are called standard levels. There are five main classes of land cover (level 1 and 2):

1. Artificial surfaces—built-up areas, including residential areas, commercial and industrial areas, mines, and green urban spaces.
2. Agricultural areas—arable land, permanent crops, meadows, pastures, and land principally occupied by agriculture with significant areas of natural vegetation.
3. Forests and semi-natural areas—forests, shrubs and open areas with little or no vegetation.
4. Wetlands—inland marshes, peatbogs, salt marshes, salines, and intertidal flats.
5. Water bodies—inland waters and marine waters.

Different land-use types within each of the above groups are specified at the second and third levels of the inventory. In the presented study, level 3 was chosen (see Table 2 in Section 2.3). The CLC uses a Minimum Mapping Unit (MMU) of 25 hectares (ha) for areal phenomena and a minimum width of 100 m for linear phenomena. The CLC vector layers were downloaded from the Pan-European datasets of Copernicus Land Monitoring Services.

**Table 2.** The values of the degree of belonging of land cover to the urban landscape.

| Land Cover according to CLC—Level 3 | Degree of Belonging of Land Cover to the Urban Landscape (M) |
| --- | --- |
| 111 Continuous urban fabric | 1.00 |
| 112 Discontinuous urban fabric | 0.69 |
| 121 Industrial or commercial units | 0.97 |
| 122 Road and rail networks and associated land | 0.82 |
| 123 Port areas | 0.82 |
| 124 Airports | 0.82 |
| 131 Mineral extraction sites | 0.64 |
| 132 Dump sites | 0.64 |
| 133 Construction sites | 0.64 |
| 141 Green urban areas | 0.68 |
| 142 Sport and leisure facilities | 0.66 |
| 211 Non-irrigated arable land | 0.09 |
| 212 Permanently irrigated land | 0.09 |
| 213 Rice fields | 0.09 |
| 221 Vineyards | 0.26 |
| 222 Fruit trees and berry plantations | 0.26 |
| 223 Olive groves | 0.26 |
| 231 Pastures | 0.09 |
| 241 Annual crops associated with permanent crops | 0.26 |
| 242 Complex cultivation patterns | 0.26 |
| 243 Land principally occupied by agriculture, with significant areas of natural vegetation | 0.09 |
| 244 Agro-forestry areas | 0.09 |
| 311 Broad-leaved forest | 0.20 |
| 312 Coniferous forest | 0.20 |
| 313 Mixed forest | 0.20 |
| 321 Natural grasslands | 0.35 |
| 322 Moors and heathland | 0.35 |
| 323 Sclerophyllous vegetation | 0.35 |
| 324 Transitional woodland-shrub | 0.35 |
| 331 Beaches, dunes, sands | 0.35 |
| 332 Bare rocks | 0.35 |
| 333 Sparsely vegetated areas | 0.35 |
| 334 Burnt areas | 0.35 |
| 335 Glaciers and perpetual snow | 0.35 |
| 411 Inland marshes | 0.35 |
| 412 Peat bogs | 0.35 |
| 421 Salt marshes | 0.35 |
| 422 Salines | 0.35 |
| 423 Intertidal flats | 0.35 |
| 511 Water courses | 0.20 |
| 512 Water bodies | 0.20 |
| 521 Coastal lagoons | 0.20 |
| 522 Estuaries | 0.20 |
| 523 Sea and ocean | 0.20 |

*2.3. Methods*

In the spatial analyses of the intensity of urban landscapes, understood in this case as land cover, the essential step is the delimitation and classification of land cover types in terms of their urban character. The problem was what types of land cover can be described as urban and what types as rural. Furthermore, how to evaluate them to describe the urban landscape intensity? Furthermore, there are some transitional landscapes, which are difficult to be definitely assigned to urban or rural character. Therefore, the authors decided to use a fuzzy theory. Fuzzy logic is applied in the cases of complex, unclear phenomena and can define and present unspecified, uncertain information [48]. This theory was used

by Biłozor et al. [25] for identification of transitional zone between urban and rural area. In this work, a classification of 24 forms of space use was proposed and the value of the degree of belonging of land cover to the urban landscape (M) has been defined for each of them. These values were adopted and adjusted to different types of land cover (Table 2). The value of the M is set between 0 and 1, where 0 indicates rural land cover, and 1 urban land cover.

The value of belonging of land cover to the urban landscape was the basis of the Urban Landscape Intensity Index (ULII). This is represented by the following equation:

$$ULII = \sum_{i=1}^{n} fiMi \tag{1}$$

ULII—Urban Landscape Intensity Index, *fi*—share of *i*-th form of land use in the area of the metropolis/municipality/hexagon, *Mi*—the degree of belonging of land cover to the urban landscape, *n*—number of forms of land cover

Analyses of the obtained ULII were conducted at three levels of research: metropolitan area, municipality and hexagon (the area of hexagon was set as 4 km$^2$). After calculation this index, at level 2 (municipalities) and 3 (hexagons), the values of ULII were divided and classified in terms of landscape types using classification proposed by Matuszewska and Będkowski [26]:

- The urban type of landscape includes units where 0.5 < ULII ≤ 1.0;
- The transitional type of landscape includes unit types where 0.3 ≤ ULII ≤ 0.5;
- The rural type of landscape includes units where 0 ≤ ULII < 0.3.

## 3. Results

### 3.1. Metropolitan Areas Level

The values of ULII calculated for each metropolitan area are low (Figure 4). The lowest urban intensity was observed for PMA and WMA (0.18). It is interesting to note that the Warsaw metropolitan area is not the most urban one in Poland. The highest values of ULII are achieved by metropolitan areas which are similar in terms of function and structure, RM (0.34) and US-ZM (0.33). MAW was in third place with much lower ULII value (0.26). The results of ULII show a connection with population density (Figure 5).

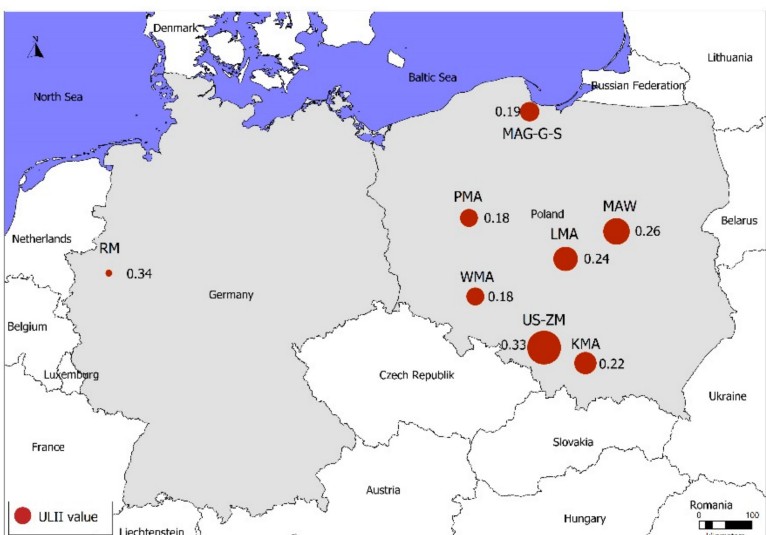

**Figure 4.** The value of urban landscape intensity indicator (ULII) in the metropolitan areas.

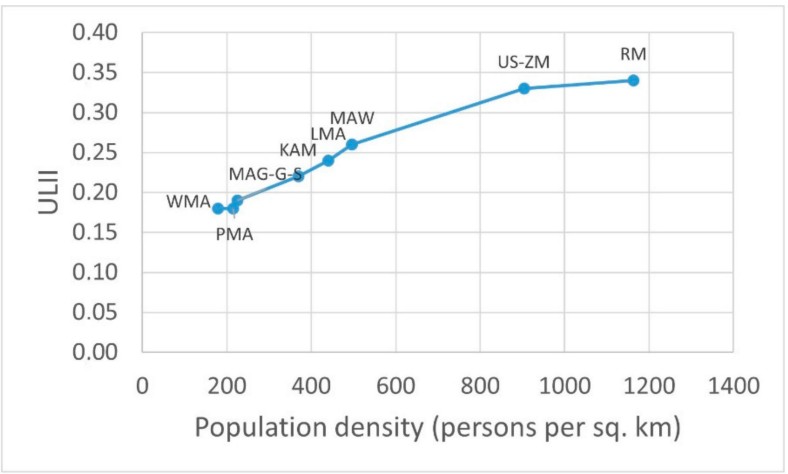

**Figure 5.** Relation between the value of ULII and population density.

*3.2. Municipalities Level*

The values of ULII at the level of municipalities show spatial differentiation. The highest ULII for a municipality varies between 0.46 (in WMA) and 0.66 (in US-ZM). It is worth noticing that in the case of the metropolitan areas with the highest ULII, i.e., US-ZM, RM and MAW, the municipalities with the highest ULII value (Świętochłowice for US-ZM, Herne for RM and Piastów for MAW) are not the biggest or main cities of the metropolitan areas (Supplementary Materials, Figure S1).

The medians of ULII in particular metropolitan areas vary between 0.17 (MAG-G-S, PMA, WMA) and 0.29 (USZM, RM) (Figure 6). The median is also high in the case of MAW (0.24). These values of median are typical of rural types of landscape. The values of ULII in the interquartile range in US-ZM, LMA, MAW and RM are typical of rural and transitional types of landscape. In other metropolitan areas, all values of ULII in the interquartile range are typical of rural types of landscape.

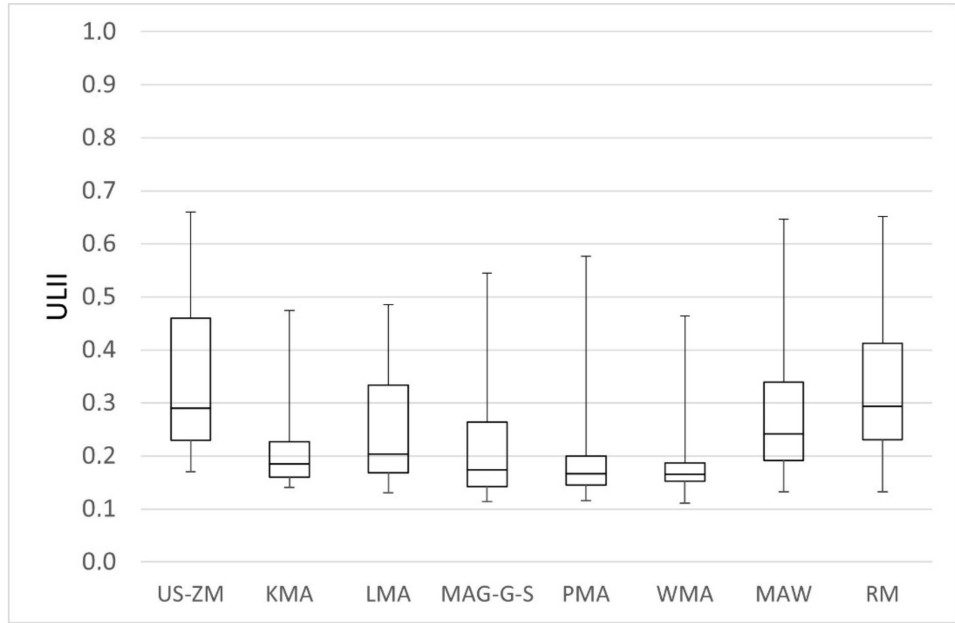

**Figure 6.** The distribution of the values of ULII on the municipalities level.

The values of ULII in the interquartile range are low. The distribution of values of ULII is characterized by right asymmetry, which means that the municipalities with lower values of ULII are dominant. The highest ULII were noted in US-ZM (0.66—Świętochłowice),

RM (0.65—Herne) and MAW (0.65—Piastów). Municipalities characterised by urban landscape (ULII > 0.5) are located in five metropolitan areas: MAW (7 municipalities: Piastów, Legionowo, Pruszków, Mińsk, Mazowiecki, Ząbki and Warszawa), RM (6 municipalities: Herne, Gelsenkirche, Oberhausen, Bochum, Essen and Duisburg), US-ZM (5 municipalities: Świętochłowice, Chorzów, Siemianowice Śląskie, Sosnowiec and Bytom), MAG-G-S (4 municipalities: Malbork, Pruszków Gdański, Puck and Tczew), and PMA (Luboń and Kościan) (Figure S1). The other municipalities are typical of transitional and rural types of landscape. In turn, the lowest ULII are 0.11–0.14. Only US-ZM stands alone in terms of having the lowest ULII value (0.17 in Zbrosławice) (Figure S1). These low values of ULII occur in municipalities located in the outskirts of metropolitan areas. The lowest values of ULII (0.11) are in Jordanów Śląski and Domaniów, in WMA, and in Lichnowy, in MAG-G-S (Figure S1).

The highest differentiation of ULII in the interquartile range occurs in RM, MAW and US-ZM. A large differentiation also occurs in LMA and MAG-G-S. The other metropolitan areas are characterized by higher similarities of ULII values in the interquartile range. The highest values of ULII in the interquartile range occur in municipalities in US-ZM and RM. MAW also stands out on this score.

In the spatial analysis of the ULII distribution for municipalities, RM and US-ZM clearly stand out. For both areas, the municipalities with high values of ULII are arranged in a wide strip with an east–west direction. The values of ULII are diversified. In addition, in MAW the municipalities are differentiated but the structure is concentric with stellar features. The highest values of ULII are located beyond the center. Other metropolitan areas also have a concentric character. For KMA, LMA and WMA, the highest ULII is present in the main cities. In WMA, the main city (Wrocław) is dominated by municipalities of a low value of ULII with the exception of Oleśnica and Oława (Figure S1). PMA and OMG-G-S also have a concentric character, but the highest values of ULII are in a few small cities spread across the area (Figure 7).

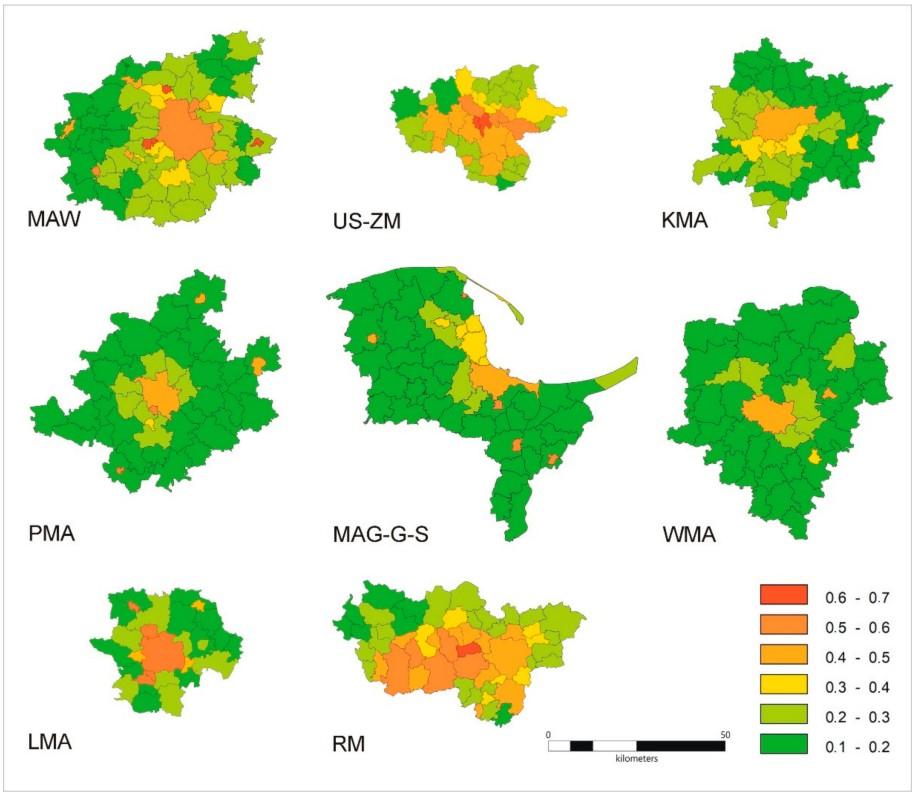

**Figure 7.** The values of ULII in the municipalities.

The spatial structure of the urban–rural area is differentiated (Figure 8). It should be emphasized that the urban type occupies small parts of metropolitan areas. The largest area of the urban type is present in RM and takes up only 18.74% of the space. Urban type municipalities form a core of 6 municipalities there (Bochum, Herne, Gelsenkirchen, Essen, Oberhausen, Duisburg) (Figure S1). US-ZM has a similar structure of types of municipalities. The urban core (9.11%) is also formed by several cities (Sosnowiec, Siemianowice Śląskie, Świętochłowice, Chorzów and Bytom, Figure S1). In the case of MAW, the urban areas of Warsaw city are almost surrounded by transitional types. In PMA, MAG-G-S, KMA, WMA and LMA, the biggest cities are of the transitional type. Moreover, in KMA, LMA and WMA the urban type of municipality is not present at all. WMA, PMA and MAG-G-S have the most rural areas (above 90% of this type). Between municipalities which form wide rural areas, there are also a few urban ones. The main city (WMA) or the main city with the neighboring municipalities (KMA and LMA) form a transition zone surrounded by a rural zone with individual municipalities of the transitional type.

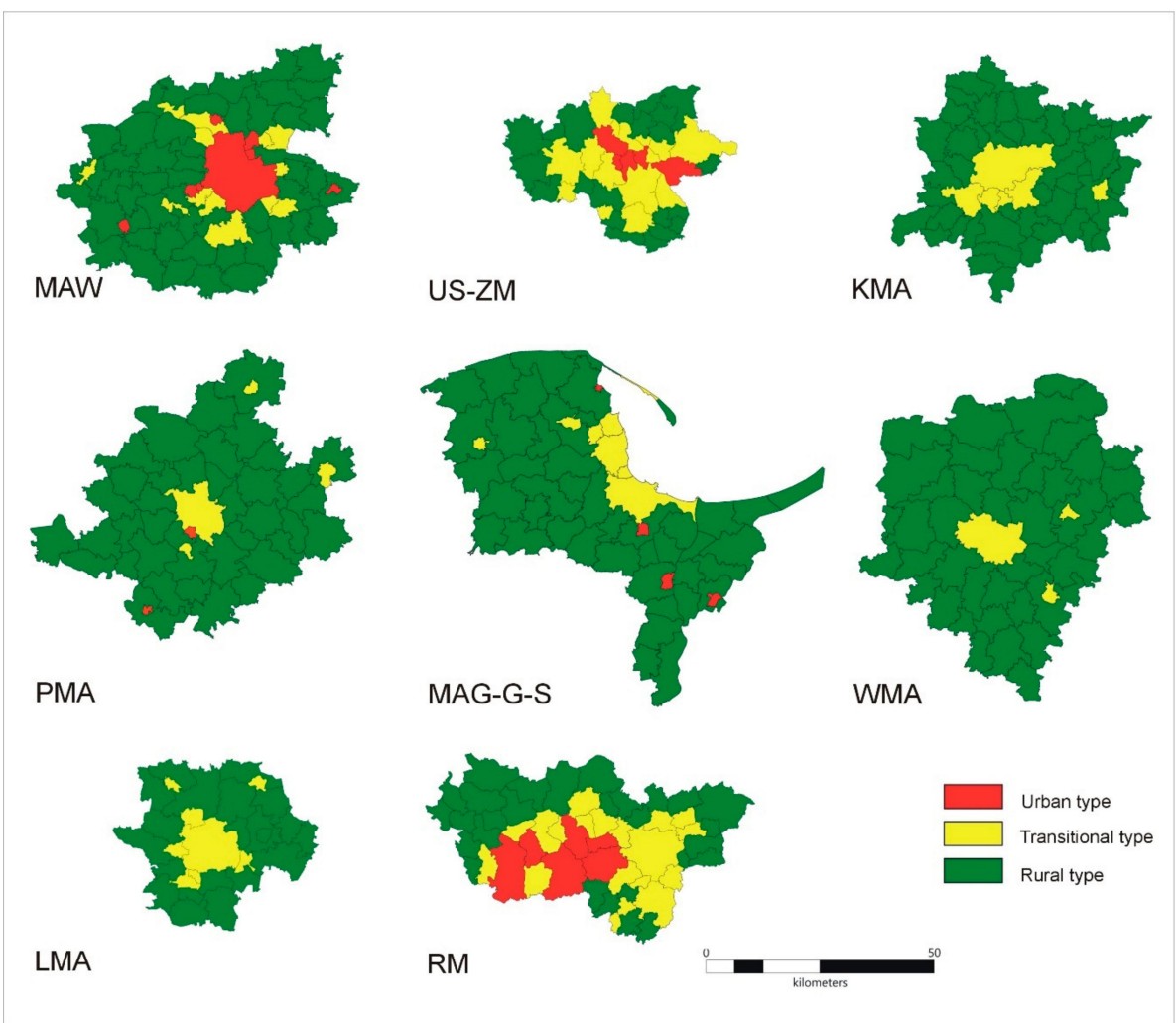

**Figure 8.** Types of landscape in municipalities in accordance with ULII.

### 3.3. Hexagon Level

The ULII for hexagons gives more detailed results. Medians of ULII in particular metropolitan areas vary between 0.15 and 0.25 (Figure 9). These median values are typical of rural types of landscape. The median in US-ZM is higher than in other metropolitan areas (by 0.05 than in MAW and by 0.1 than in PMA). Simultaneously, this is the same

value as in RM. The values of ULII in the interquartile range only in US-ZM and RM are typical of rural and transitional types of landscapes. In other metropolitan areas, all values of ULII in the interquartile range are typical of rural types of landscape.

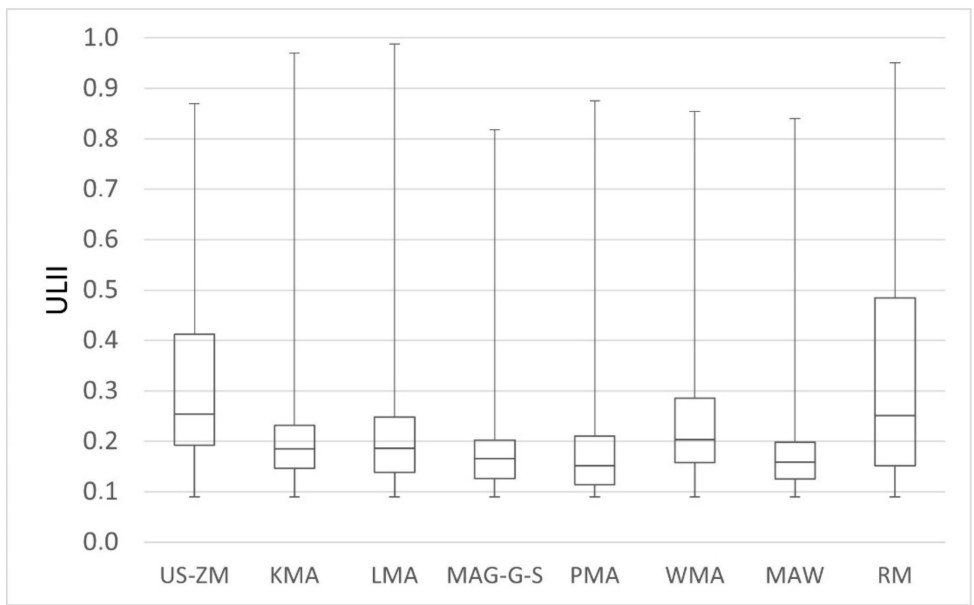

**Figure 9.** The distribution of the values of ULII on the hexagons level.

The values of ULII in the interquartile range are low. The distribution of values of ULII is characterized by right asymmetry, which means that the hexagons with lower values of ULII are dominant. The highest ULII were noted in LMA (0.99) and KMA (0.97). These hexagons occur in midtown in Łodź, where continuous urban fabric and road and rail networks and associated land dominate, and in Kraków Nowa Huta where there are industrial or commercial units. Simultaneously, these metropolitan areas are characterised by the highest differentiation of ULII. In the case of RM, the maximal ULII is 0.95 in Essen, where industrial or commercial units dominate and continuous and discontinuous urban fabric, road and rail networks and associated land, and green urban areas occur. This value is higher than in US-ZM, where the maximal ULII is 0.87 in Katowice-Bogucice, where industrial or commercial units occur, but there are also broad-leaved forests. In turn, the lowest ULII in all metropolitan areas is the same (0.09). This concerns the outskirts of metropolitan areas where non-irrigated arable land predominates and, in some parts, pastures occur.

The highest differentiation of ULII in the interquartile range occurs in RM, US-ZM and MAW. The other metropolitan areas are characterized by a higher similarity of ULII values in the interquartile range. The highest values of ULII in the interquartile range occur in RM and US-ZM. MAW also stands out on this score. RM is characterized by the highest differentiation of ULII in the interquartile range, which is similar to US-ZM. The maximal values of ULII in the interquartile range in RM are higher than in US-ZM. However, the lowest values of ULII in the interquartile range are lower than in US-ZM.

Spatial analyses show the concentration of high values of ULII in the center of the metropolitan area in KMA, PMA, LMA, WMA (Figure 10). In KMA, the structure of types of hexagons is evenly concentric with a few exceptions. The intensity pattern of the ULII related to the layout of the main roads is visible there. PMA forms a concentric character, stretching to the east (with Swarzędz and Kościan, Figure S1) with numerous small isolated, spotted, scattered high value units and a stronger satellite formed by Gniezno (Figure S1). In LMA there is a clear concentration of the highest values of the ULII in the center of Łódź. High values are also reported in Pabianice, Zgierz and Koluszki (Figure S1), forming a triangular system related to the railway line. In WMA, the main city (Wroclaw) is

surrounded by municipalities with a low value of ULII. A few small isolated, spotted, scattered high value units are present, and a small satellite is formed by Oleśnica (Figure S1). In the case of MAW, the structure is stellar and high ULII values are present along the main roads leaving the city. In MAG-G-S, a concentration of high values of ULII is visible along the coast on the line Gdansk-Gdynia-Sopot, with Gdynia taking on a more concentric character, and Gdansk having a more even structure. US-ZM and RM urban areas form a wide strip in an east-west direction. In US-ZM, the core is more compact, while in RM the highest value of ULII is more dispersed.

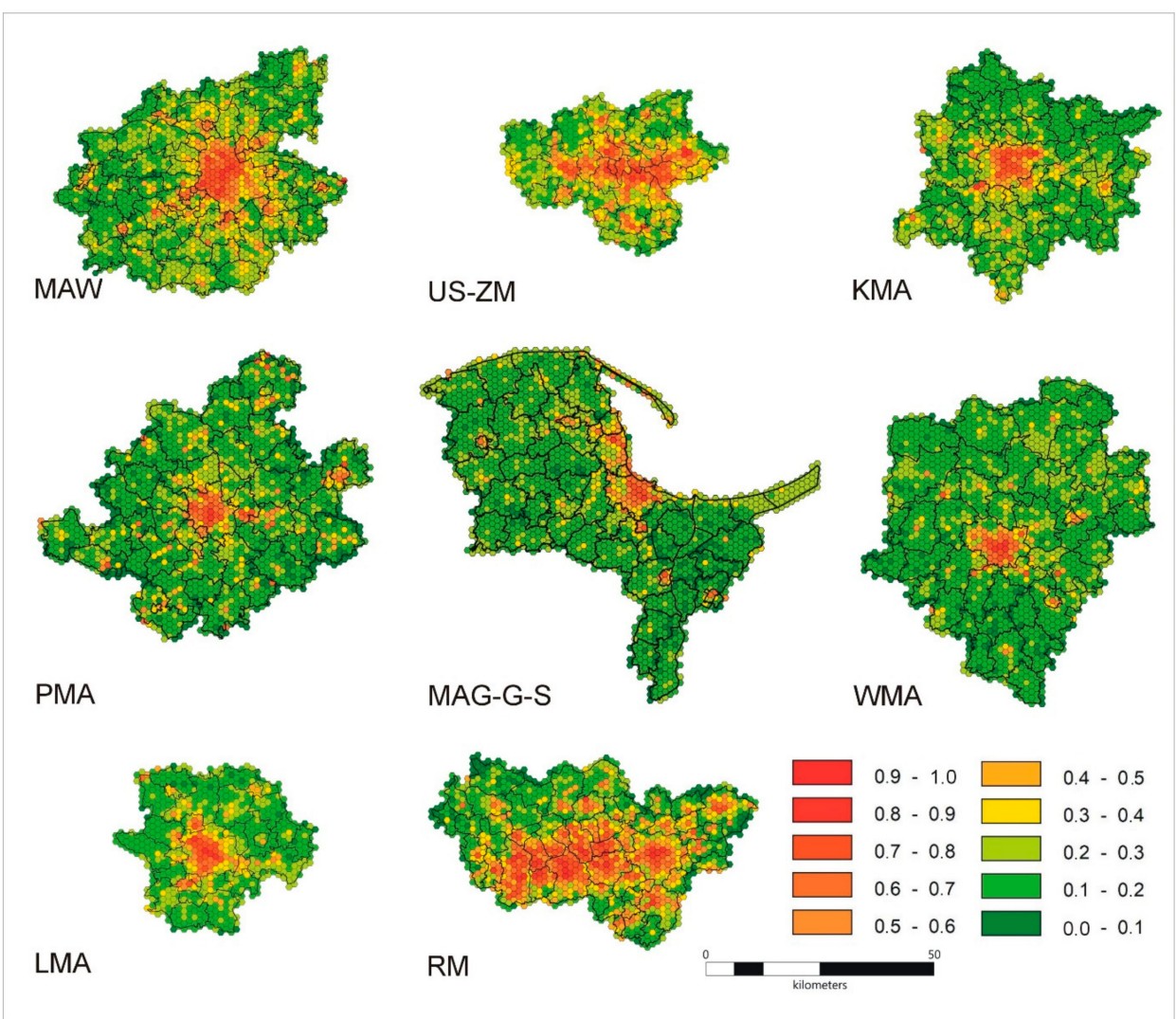

**Figure 10.** The values of ULII in the hexagons.

Spatial analyses of the urban–rural type show the domination of the rural type in most of the research areas (Figures 11 and 12). This domination is especially visible in WMA, where the urban type is concentrated in a small area of the main city with a thin border of the transitional type. Outside the main city, the urban and transitional types are present in a few small spots. A similar situation is observed in MAG-G-S, with urban types located in three main cities and single, dispersed, isolated urban hexagons located in the whole area. Often, they are related to tourist functions (Łeba, Władysławowo, Jastarnia, Kuźnica, Chałupy). It is significant that, administratively, Chałupy and Kuźnica are villages. A smaller difference in proportions between urban and rural areas is visible in KMA, LMA and PMA. In PMA, the urban type is concentrated in Poznań, but there are also numerous urban and transitional units dispersed all over the area with a more

concentrated spot in Gniezno. In KMA, the urban type of Krakow is surrounded by the transitional type. In the east, Bochnia has a characteristic concentric arrangement of urban and transitional types. In LMA, the urban type located in Lodz has a wide border of transitional units. Urban units are also present in Pabianice, Zgierz and the more isolated Koluszki. For US-ZM, a compact east-west urban core spills over into Tychy, Knurów, Ożarowice and Tarnowskie Góry (Figure S1). The largest share of urban type units is present in RM. The urban core forms a wide strip of units located in several municipalities between rivers: Emsher and Ruhr (i.e., Dortmund, Bochum, Essen, Oberhausen, Herne, Gelsenkirchen and Gladbeck, Duisburg). Numerous urban units are also present in Hamm, Hagen and Marl (Figure S1).

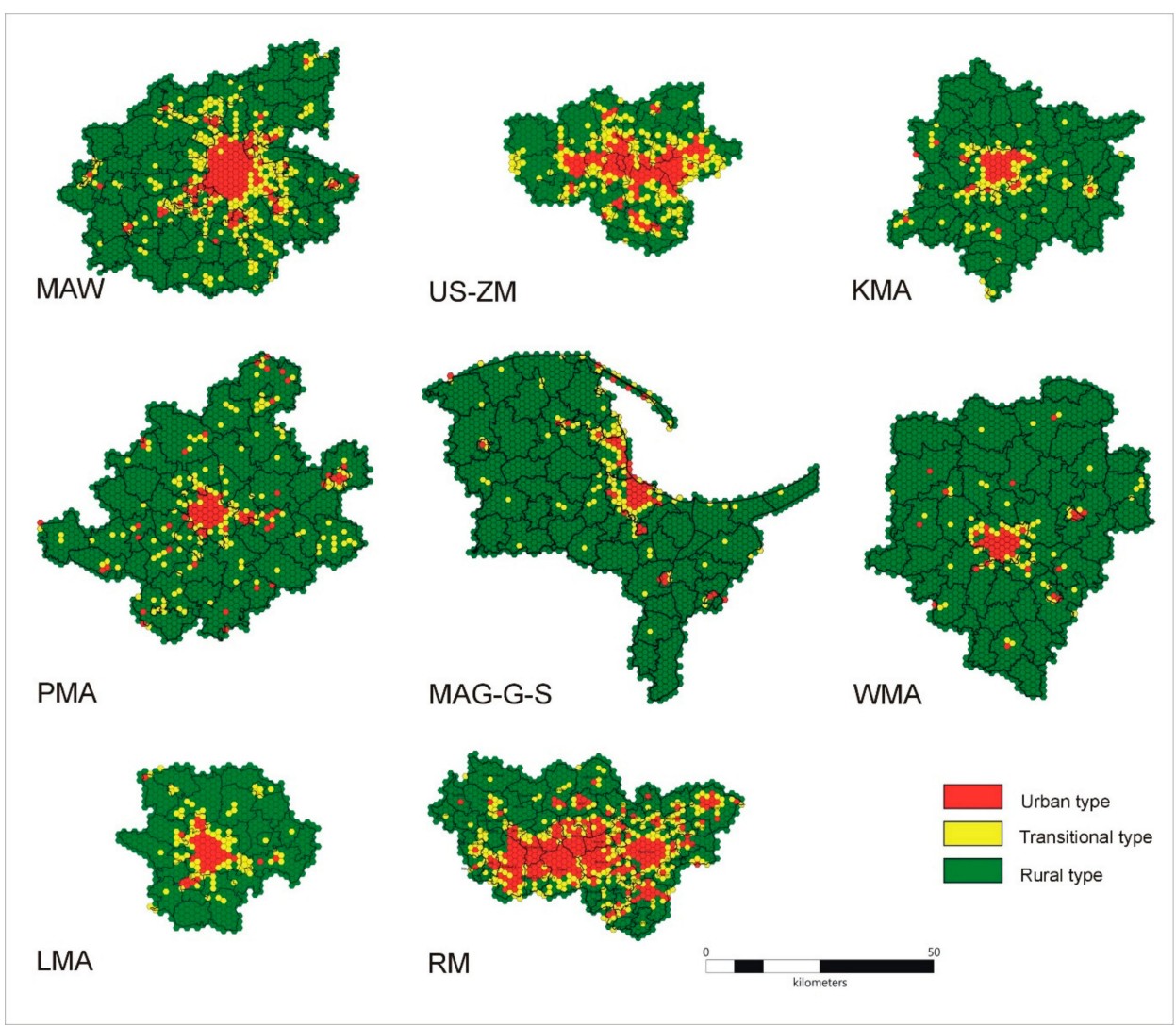

**Figure 11.** Types of landscape in hexagons in accordance with ULII.

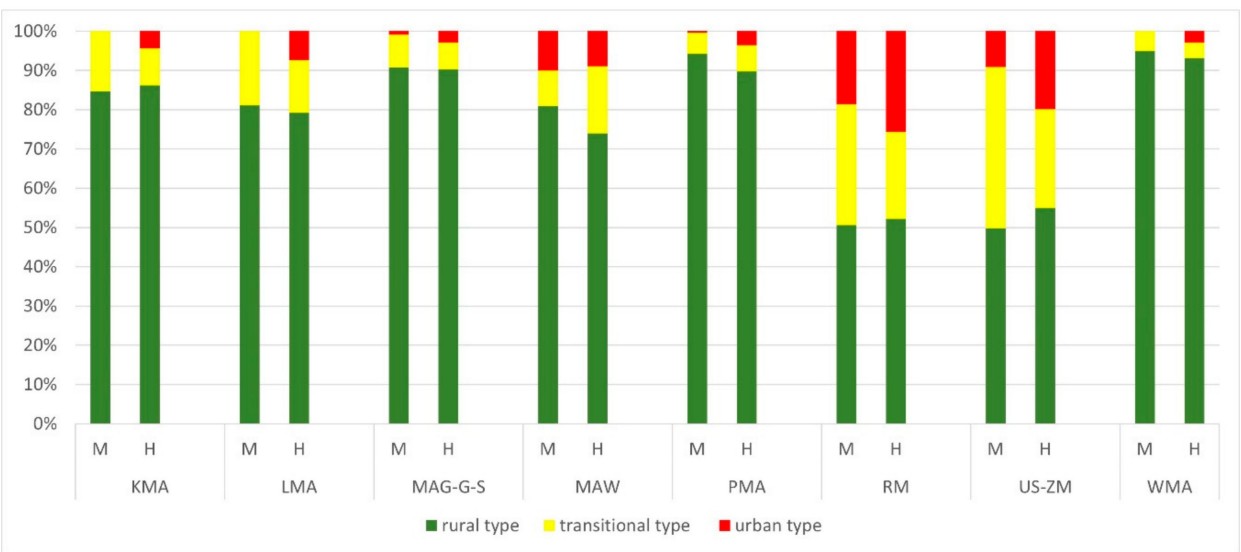

**Figure 12.** Landscape types in municipalities (M) and hexagons (H).

## 4. Discussion

In this paper, the intensity of the urban landscape for seven metropolitan areas in Poland was assessed. Additionally, the landscape of RM in Germany was examined as a comparative area for US-ZM—the only polycentric agglomeration in Poland. The use of the ULII index showed that the landscape of the US-ZM differs from the landscape of other metropolitan areas in Poland (higher average values of ULII). At the same time, the landscape of the US-ZM shows some similarities to the landscape of RM (similar values of ULII). Moreover, MAW also shows some similarities to the US-ZM in terms of landscape diversity. What is more, the study showed that urban landscapes occur only in the central zones of the metropolitan areas that were delineated on the basis of official documents. In turn, rural and transitional landscapes predominate as far as percentage of their surface is concerned.

The similarity of the US-ZM and RM landscapes may result from the similar factors that influenced development of these metropolitan areas. In both cases, the mining industry was a factor in the location and development of cities [49]. Moreover, the spatial nature of the agglomeration, polycentric in the case of US-ZM and RM, and monocentric in the case of other metropolitan areas, has an influence on the landscape differentiation and similarities which occur [50]. The value of ULII could also be influenced by a degree of industrialization. High values of ULII occur in former industrial districts such as Nowa Huta (Kraków), Bałuty, Fabryczna (Łódź), Psie Pole, Kowale (Wrocław) and Młyniska (Gdańsk).

The value of ULII could be influenced by many different factors. The initial comparison of the spatial distribution of ULII indicates that among these factors transport network systems and their density, and the spatial organization of metropolitan areas (polycentric or monocentric) can be distinguished [51]. The occurrence of raw materials is also of great importance as they stimulate the economic development of metropolitan areas (this is in the case of US-ZM and RM) [52], and their primary and secondary functions [53,54]. Furthermore, the natural environmental conditions are crucial [55]. They may be favorable (hydrographical network, access to the sea, favorable topoclimatic conditions) or unfavorable (high mountain areas, boggy areas, proximity of protected areas) for the spatial development of metropolitan areas [56–58]. The crucial factor that should be taken into account when interpreting the value of ULII is the process of changes on the border of metropolitan areas in official administrative documents. This process is connected with decisions about incorporating particular municipalities that are of different types: rural, urban or urban–rural. Furthermore, economic potential [59], spatial policy and land

prices [60–62] may have an influence as well. Many aspects of urbanization are interpreted through globalization [63].

Nevertheless, the most visible aspect is the relations between the values of ULII and transport network systems and industrial districts. However, this issue requires further recognition in our study area through the statistical analysis of the correlation that occurs between these variables and the construction of a regression model that would allow the occurring relationships to be quantified. Such analyses were conducted by Conway and Hackworth in the Greater Toronto Area [34]. A detailed analysis of landscape urbanization in the context of driving forces in our study area will be the subject of a separate study by the authors. Nevertheless, some basic conclusions could be drawn. The analysis of the hexagons level reveals a star-shaped pattern of the urban landscape in MAW. Additionally, in US-ZM and RM the intensity of the urban landscape is visible in a linear pattern. This is due to the emergence of the urban sprawl process along the transport network which is typical of other metropolitan areas [64–66]. In MAW, the urban landscape is visible along the A2 motorway, and the S8 and S7 expressways along the east-west and north-south routes. The main railway lines to the capital run parallel to the road network. In US-ZM and RM, the intensity of the urban landscape increases along the east-west line, which is connected with the railway between Gliwice and Mysłowice and the A4 motorway in Poland, and the A2 and A42 motorways in Germany. The relationship between the development of the road network and the processes of suburbanization has been confirmed by Garcia-Lopez [67] who, based on the example of Barcelona, stated that the construction of new road infrastructure generates urban sprawl processes. Baum-Snow [68] puts forward the thesis in a similar way, examining the relationship between suburbanization and road development in the United States. Moreover, it is worth noting that the ULII is influenced by the types of urban coverage, which include residential buildings related to urban sprawl processes, as well as industrial areas and large-surface communication junctions. Thus, the transport network influences the level of ULII both directly and indirectly.

Similar research has already been carried out in Olsztyn and Sieradz (Poland) [25,26]. The areas classified as urban in Olsztyn cover only 24% of the city's area. In turn, in Sieradz non-urban areas also prevail. The percentage of agricultural areas within the boundaries of metropolitan areas in Poland was studied by Sroka et al. [36]. Their study shows that 49.9% of the areas administratively belonging to metropolitan areas are occupied by farms. Similar comparative studies conducted for RM and US-ZM showed that agricultural areas cover 39.2% of the RM area and 42.7% of the USZM [38]. These values are lower than the results presented in this paper (81% of seven metropolitan areas in Poland are covered with rural landscapes, 52% in RM, and 55% in US-ZM, respectively). However, the differences result from the different classifications of rural landscapes. In this paper, rural landscape includes arable land, pastures and forests. Hence, this landscape type in our paper has a broader scope. The applied approach allowed the assessment that only in the US-ZM and RM is it possible to distinguish the core of a metropolitan area, in which urban landscapes have the largest percentage among all the analyzed areas. A comparison of landscape types in municipalities and hexagons shows differences in the accuracy of the results. In accordance with the adopted criteria, there are no urban landscapes in KMA, LMA and WMA at the level of municipalities, while at the level of hexagons their occurrence is visible. On average, the share of rural and transitional landscapes is higher at the municipality level than at the hexagon level. Hexagons illustrate with greater accuracy the spatial distribution of landscape types enabling the justification of the existing layout of urban landscapes and further spatial analyses, e.g., related to the study of urban development in terms of driving forces. Moreover, research at the hexagon level is objective and based on mathematical logic, as opposed to the level of boundaries of municipalities and metropolitan areas, whose scope is determined administratively.

As already mentioned in the introduction, many authors use the CORINE database for landscape analysis [30–32]. Nevertheless, it is worth noting that despite many advantages, this database has some limitations [69]. However, this database is imperfect in small-scale

studies, while in landscape analysis it is an appropriate source [33]. Admittedly, there are many other databases that may be used in such studies. For instance, in Europe, higher resolutions have Urban Atlas maps—a project developed as part of Global Monitoring for Environment and Security. Nevertheless, maps are created only for selected areas around large cities, so they do not cover the entire EU territory [69]. Another example of the increase in the spatial resolution are the "fourth-level" CLC maps, however, they were prepared only in some countries and classifications used in them are inconsistent with each other. This paper confirms the statement that the CORINE database is useful in landscape analyses, both in relation to administrative units (municipalities, metropolitan areas) and geometric basic fields. It is a particularly reliable source in studies of urban areas undergoing constant change [31]. It must be emphasized that the value of ULII could be influenced by many different factors e.g., different types of a dataset, map scale, type of basic units (their size and shape). While using ULII these limitations must be taken into account.

## 5. Conclusions

In this paper the intensity of the urban landscape for seven metropolitan areas in Poland and one in Germany was assessed. The analyses were conducted at three levels of detail: the metropolitan area level, municipalities level and hexagons level. The conducted research allows the following conclusions to be drawn:

1. The landscape of the US-ZM differs from the landscape of other metropolitan areas in Poland and shows some similarities to the RM landscape. These areas are polycentric agglomerations. Nevertheless, the landscape of MAW shows some similarities to the US-ZM one despite the fact that this area is a monocentric agglomeration. This may be connected with the capital function of Warszawa that has an influence on the urbanization level.
2. Rural and transitional landscapes predominate within the metropolitan areas delineated according to official documents and administrative affiliation, while the typical urban landscape is characteristic only of the central zones of metropolitan areas. This conclusion may be considered controversial because it proves that in determining the features of a metropolitan area the landscape approach differs from the formal and legal approaches and also based on area and demographic criteria.
3. Basic landscape analysis on an administrative level gives a distorted spatial picture. Therefore, the use of basic units (in this case, hexagons) for the assessment of metropolitan landscape diversity is recommended. Only by using this analysis can all the complexity and diversity of the internal structure of the metropolitan areas landscape be shown.

The conducted studies give new research prospects concerning landscape analysis in metropolitan areas such as driving forces. Furthermore, the changes in ULII value in particular time periods may be also analyzed. Nevertheless, it must be emphasized that the same dataset must be used to make the results comparable. It is connected with the fact that the value of ULII could be influenced by many different factors such as different types of dataset, map scale, type of basic units. Monitoring landscape changes based on the CLC update and the use of the indicators presented in this article, and predicting the directions of landscape transformation as a result of driving forces are some of the most important challenges of contemporary interdisciplinary research. The results of the research may be useful for regional policy, e.g., in the preparation of urban planning documents and spatial development strategies.

**Supplementary Materials:** The following are available online at https://www.mdpi.com/2073-445 X/10/1/51/s1, Figure S1: Administrative division of metropolitan areas.

**Author Contributions:** Conceptualization, U.M.-P.; methodology, A.Ż.-S., K.P.-K. and M.S.; software, J.N.; formal analysis, A.Ż.-S., K.P.-K. and M.S.; investigation, A.Ż.-S., K.P.-K. and M.S.; resources,

J.N.; writing—original draft preparation, U.M.-P., A.Ż.-S., K.P.-K. and M.S. All authors have read and agreed to the published version of the manuscript.

**Funding:** This research received no external funding.

**Institutional Review Board Statement:** Not applicable.

**Informed Consent Statement:** Not applicable.

**Data Availability Statement:** Publicly available datasets were analyzed in this study. This data can be found here: Corine Land Cover: https://land.copernicus.eu/pan-european/corine-land-cover, Chief Geodesy and Cartography Office (GUGiK) data: https://mapy.geoportal.gov.pl/imap/Imgp_2.html, OpenGeodata of Nord-rhein-Westwalen data: https://www.opengeodata.nrw.de/produkte/ (accessed on 19 November 2020).

**Acknowledgments:** We gratefully acknowledge three anonymous reviewers for their constructive comments.

**Conflicts of Interest:** The authors declare no conflict of interest.

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
