# Peer review of "Is There Urban Landscape in Metropolitan Areas? An Unobvious Answer Based on Corine Land Cover Analyses"

_land, doi:10.3390/land10010051_

Round 1

Reviewer 1 Report

Thank you for the opportunity of reviewing the paper entitled:''Is there Urban Landscape in Metropolitan Areas? An Unobvious Answer Based on Corine Land Cover  Analyses''.

The paper aimed to compare the landscape  diversity of the seven largest metropolitan areas in Poland,by  using the urban landscape  intensity index (ULII).

My comments/suggestions are:

1.The introduction is the synthesis of the whole work. Therefore the authors must present briefly the main results, thus emphasizing their contribution.

I think the statement on line 115:''Such an  approach will guarantee the sustainable planning and management of constantly growing urban  areas'', is hazardous.

2.The value of ULII could be influenced by many different factors. Limitation of the study is necessary,while using other evaluating factors,diffrent types of dataset etc, the analysis may conduct to diffrent results.

3.In addition,the authors should outline their contibution and show the novelty of their research in the context of previous works and results obtained. This is important because thus authors could demonstrate what added value bring their research to the area of knowdlege.

4.The source of data and software used should be attached to the figures and tables.

Reviewer 2 Report

Summary The authors use CORINE Land Cover (CLC) inventory data to calculate an Urban Landscape Intensity for seven metropolitan areas in Poland and one in Germany. ULII is calculated at three administrative levels and comparisons made across the cities. The study found 'urban' landscapes located in cores of metropolitan areas whereas 'rural' and 'transitional' landscapes were dominant.   I found the literature review wanting in that it did not sufficiently introduce the methods used in this study or provide a justification for the study. The authors should very early on in the paper define the meaning of 'urban' and 'rural' landscapes and describe the various units of urban level analysis adapted in the study. The methods were also convoluted and difficult to follow. The discussions were quite lengthy and could benefit from being made more concise. There are many locations mentioned that are not found in any of the maps , thus reducing their value in the discussions.   Below I provide additional observations and comments.  

Abstract

Line 20: Let readers make the call whether findings are 'interesting'.  

Introduction

Line 29: Avoid using gendered nouns such as 'mankind' and replace with gender neutral ones such as humankind, humans' humanity, human beings, etc.  

Lines 90-91: Change to past tense to maintain consistency in the first sentence. Modify beginning of second sentence here as 'In this work .......'. The sentence need elaboration - it would be useful to inform viewers the key aspects of the CORINE land cover model without having to force readers to read the cited paper just to get these basics.  

Lines 91- 92: A model is not a 'material'. The sentence needs to be elaborated. The model was used as a basis for what? In the next sentence , still in reference to CORINE, you refer to a 'database', which I assume is in reference to CORINE. It would be much more informative to readers if you began the paragraph starting in line 90 with a brief description of the CORINE land cover inventory program, since this is the basis of the research described in this manuscript. Although the CLC inventory is described to some detail in later sections (starting in Line 145), it should be outlined here briefly.  

Line 96-98: I am quite puzzled by the questions posed here. There is nothing in the literature described up to that point to suggest that those questions cannot be answered using CORINE land cover inventory. There are a dozen of studies that have answered similar questions to these that used databases of that included various built-up classes , e.g., Conway and Hackworth, 2007, Dadashpoor et al., 2019; Inostroza et al., 2019; Bian et al., 2021.  

Lines 99-100: Make sentence more concise by beginning sentence as 'The aim of this study is to compare.....' I would suggest the authors review additional literature to justify the study goal of comparing landscape diversity in metropolitan areas in Poland.  

Lines 175-183: The section beginning here needs to be rephrased for grammar and clarity. For example, sentences in lines 176-178 are vague. Explanations need to be complete and following a logical sequence. Begin by stating that the metropolitan areas were selected based on population and describe specifically what analysis of land cover were conducted. What do you mean by 'the degree to which the land cover belongs to the urban landscape'? You next mention calculating an 'urban landscape intensity index. This should have included in the literature review in order to justify its use here. You also need to explain the various urban administrative units that are used in this analysis prior to describing the types of analysis undertaken at the different levels. Is there an urban hierarchy for the structural units? Do not assume that readers will know these urban structures. For example, what are the hexagons? What are the differences between metropolitan and municipal levels?

Lines 181-184 is yet another example of a sentence whose meaning is unclear. This needs to elaborated and rephrased to correct grammar and improve clarity.  

Lines 186-188: Not clear.

Lines 188-191: A literature search on Urban Landscape Intensity Index (ULII) hardly returns any results making it essential that you describe its use in the three studies cited more detailed. This needs to be elaborated and moved to the introduction so when it is mentioned in the methods as a major measure of analysis, the reader is not surprised.  

Line 218-219: How can you describe the relationship between population and ULII as strong when you do not calculate a measure such as correlation to quantify the relationship?

Lines 224-226: So if not in these areas, where? You do not answer this question until Lines 237-242.  

Lines 377-379: The authors should take a look at the study by Conway and Hackworth (2007) who examine relationships between urban pattern and land cover in the Greater Toronto Area.  

Lines 412-415: In what ways is the analysis at the hexagon level better? Lines 415-416: What do you mean by 'reflecting better administrative decisons...' Elaborate and provide examples of this.  

Line 443: What problem?

Lines 444-445: What would be the benefit of further comparisons? What additional insights would be gained that have not been revealed in the current study?  

The conclusions need to be more on the implications of the study, a reflection of study limitations, and recommendations for future studies.

Reviewer 3 Report

After careful check, I found that authors did a lot of work to explore whether there is urban landscape in metropolitan areas or not, based on CLC database. The whole text is very readable and fluent in English, and logical. If there were not vital weaknesses, I think the current manuscript is very well. However, I have to ask you to reconsider how use the data at your hand to do another work.

The main issue is that according to Figure 2 (L143) except US-ZM other metropolitan areas are main dominated by rural areas or urban-rural zones. In this case, these metropolitan areas what you referred to are already not real metropolis. This can also be confirmed by what you said in L66, that is, in Poland, the processes of landscape metropolization are in the initial stages. Accordingly, the values of ULII calculated for each metropolis are low (Figure 4) (L214) is obvious, and of course you have this question: in the cities one can speak of an urban landscape, but is this really evident?

Other main issues including:

1 L99: By posing the research question in this way, the aim of this article is to compare the landscape diversity of the seven largest metropolitan areas in Poland. In fact, only ULII was compared at three levels through the whole text and landscape diversity comparison among metropolitan areas was just put forwarded in conclusion section.  

2 L100-107: Without analysis, how did you know that US-ZM and RM had the similar landscape? L120: the largest seven metropolitan areas in Poland were chosen, according to the classification of urban centers in Poland based on the criterion of population. Here, you said metropolitan areas are based on population, but in Table 1, population in US-ZM is 2.3 mln, and in RM is 5.1 mln, then they are similar? In addition, polycentric and monocentric agglomerations should be relatively specific.

Round 2

Reviewer 3 Report

Thanks for your response. I still believe that this wrok has little meaning.

  1. Before L 110, no 'metropolitan area' was found. Is there a common definition of it? 
  2. L 110 the intensity of the urban landscape could be based on metropolitan areas? urban landscape is at city level, but metropolitan includes rural and urban-rural landscapes as you already pointed out. Are they the same? 
  3. according to Figure 3, the results are already known. It is necessary to be study again?
  4. in your all examples of studies on land cover and changes, only two is related to metropolitan areas in Poland (references of 28 and 29). Because I can't download these papers,  I don't know in thier paper they studied 'metropolitan areas' or just 'metropolises'.
